# Initiation of Direct Shoot Organogenesis in Coconut Using Immature Inflorescence

**DOI:** 10.3390/plants14203123

**Published:** 2025-10-10

**Authors:** Eveline Y. Y. Kong, Julianne Maree Biddle, Sisunandar Sisunandar, Sundaravelpandian Kalaipandian, Amirhossein Bazrafshan, Zhihua Mu, Steve W. Adkins

**Affiliations:** 1School of Agriculture and Food Sustainability, The University of Queensland, Gatton Campus, Lockyer Valley Region, Gatton 4343, Australia; julianne.biddle@uq.edu.au (J.M.B.); s.kalaipandian@uq.edu.au (S.K.); amirhossein.bazrafshan@gmail.com (A.B.); zhihua.mu@uq.net.au (Z.M.); s.adkins@uq.edu.au (S.W.A.); 2Queensland Alliance for Agriculture and Food Innovation (QAAFI), Centre for Horticultural Science, The University of Queensland, Indooroopilly, Brisbane 4068, Australia; 3Biology Education Department, Muhammadiyah University of Purwokerto, Purwokerto 53182, Indonesia; sisunandar@ump.ac.id; 4Department of Bioengineering, Saveetha Institute of Medical and Technical Sciences (SIMATS), Saveetha School of Engineering, Chennai 602105, TN, India; 5Academy of Agriculture and Forestry Sciences, Qinghai University, Xining 810016, China

**Keywords:** cloning, coconut, inflorescence, direct organogenesis, thidiazuron

## Abstract

Coconut (*Cocos nucifera* L.) is one of the most widely cultivated crops, with increasing popularity and demand for its products, which necessitates increased production. However, the lack of high-quality planting materials is a major limitation in replanting the senile palms worldwide. This study examined the possibility of using a direct shoot organogenesis pathway as an alternative to somatic embryogenesis to produce clonal coconut plantlets using immature inflorescence explants obtained from Indonesia and Australia, through investigation of the explant types, exogenous plant growth regulators, and additives. Histological analysis showed suitable stages of immature inflorescence explants to be used, which led to the formation of shoot-like structures resembling true vegetative shoots, in all treatments consisting of exogenous plant growth regulators except for those without activated charcoal. The culture medium supplemented with thidiazuron (100 μM) alone or those supplemented with various combinations of other plant growth regulators showed similar shoot induction percentages (ca. 63 to 80%) or shoot-like structures per explant (ca. 6 to 8). The addition of adenine sulphate (217 μM) was found to significantly improve shoot induction (ca. 50%) from immature inflorescence explants compared to the control (ca. 5%), whereas phloroglucinol was found to negatively impact shoot induction, and L-glutamine showed a positive influence. The current study showed several improvements, which warrant further studies to develop commercial protocol for mass production of clonal coconut plantlets through direct organogenesis.

## 1. Introduction

Coconut (*Cocos nucifera* L.) somatic embryogenesis (SE) was first attempted by Eeuwens and Blake [1] and, for the past 60 years, has been considered the only pathway for the clonal propagation of this species [2]. This is because the coconut palm only produces a single stem and does not produce branches or suckers [3]. This single-stem characteristic is a common trait for most palm species [4], but occasional cases of shoot branching do occur [5,6]. This rare feature can lead to palms having multiple stems. Therefore, direct shoot organogenesis may be possible under appropriate conditions and could be an alternative micropropagation approach to coconut SE, as shown in oil palm (*Elaeis guineensis* Jacq.) and date palm (*Phoenix dactylifera* L.) [7]. Direct shoot organogenesis generally offers a more secure regeneration pathway because it reduces the chances of somaclonal variation occurring, as an intervening callus phase is not required [8]. However, it is a poorly studied pathway in comparison to SE for coconut, hence only a few studies in the literature exist to date [3,9,10,11]. From the previous studies, only a few shoots have been regenerated, and the difficulty in obtaining plantlets suggests that further research is needed to improve direct organogenesis pathway for coconut.

Direct shoot organogenesis in palms has been shown to occur from zygotic embryos, shoot tips, immature inflorescence, and shoot tip organogenic clusters. A preliminary study on an in vitro shoot multiplication protocol via. organogenesis from the apical meristems of coconut seedlings has been recently undertaken [3]. Exogenous plant growth regulators (PGRs) have shown an important role in coconut organogenesis [3,9,10,11,12], as they can promote shoot formation and development through de novo establishment of polar auxin gradients and their multidirectional transport [13]. Cytokinins, alone or in combination, with auxins are the most efficient PGRs for stimulating shoot organogenesis [14]. According to Chandrakala, Renuka, and Sushmitha [11], 150 µM thidiazuron (TDZ) with activated charcoal (AC) promoted coconut shoot organogenesis (75% shoot initiation, 66% shoot regeneration, maximum two shoots per mature zygotic embryo), but inhibited when 2,4-dichlorophenoxyacetic acid (2,4-D) was added to the treatment. Similarly, two shoots were induced from a single Kopyor coconut zygotic embryo using kinetin (10 µM) and indole-3-butyric acid (2–4 µM) [15]. Another synthetic cytokinin, 6-(γ,γ-dimethylallylamino)purine (2iP; 9.8 µM) without AC in the medium, promoted direct shoot organogenesis of oil palm to produce a 54% shoot induction rate with five shoots being produced per responsive shoot explant [16]. In date palm, the combination of several exogenous cytokinins, auxins, and an auxin uptake inhibitor applied at a low concentration was found to induce shoot formation, but the rate at which this occurred depended on the cultivar studied [7].

Coconut immature inflorescences remain the preferable explants due to their ease of collection and the ability of somatic tissues to produce true-to-type clones, when compared to the shoot tips coming from zygotic sources [17]. Immature inflorescence explants from coconut have been shown to produce shoots directly when placed on a medium supplemented with low concentrations of 6-benzylaminopurine (BAP) or TDZ, and in combination with auxins (2,4-D or picloram; PIC), in the presence of AC [9,10]. However, coconut inflorescence explants cultured on Eeuwens Y3 [18] (Y3) medium, containing a combination of BAP and 2,4-D, only produced a low frequency of shoot production [9]. In another study, shoots were formed from immature inflorescence explants of coconut under diffused light at a rate of 1 to 14 shoots per explant when cultured on half-strength Murashige and Skoog (MS) medium [19] supplemented with 8.3 µM PIC and 4.5 µM TDZ. However, a combination of 2,4-D and TDZ resulted in callus formation on the coconut inflorescence explants [10]. Similarly, the number of shoots produced from West Coast Tall immature inflorescence explants cultured on different media and PGRs at different stages also varied from 1 to 17 shoots after 12 to 14 months [12].

To enhance the rate of organogenesis from immature inflorescence explants, other additives such as adenine sulphate (AdS or adenine hemi-sulphate), phloroglucinol (PG), and L-glutamine may be used to promote shoot induction in coconut. Adenine sulphate may enhance plant growth by acting as a substrate for the biosynthesis of endogenous cytokinins that improve plant growth through a feedback inhibition process [20], leading to growth-promoting properties. The effect of AdS alone on shoot proliferation was weak in comparison to other cytokinins [such as BAP or kinetin (KIN)] in a study on Java plum (*Syzygium cumini* L.), but it could enhance shoot proliferation by reinforcing the effect of other PGRs [21]. The synergistic effect of AdS with other cytokinins has been shown to promote multiple shoot formation and proliferation in the physic nut (*Jatropha curcas* L.) [22]. Although there have been no studies so far on coconut, AdS (ranging from 108.6 to 434.4 µM) has been added to the culture medium to promote organogenesis in both shoot tip and immature inflorescence explants of date palm [7]. Hence, as adenine is a synergist of cytokinins [20], the addition of AdS is likely to improve immature inflorescence organogenesis of coconut, as it did for date palm.

Phloroglucinol, is a polyphenolic compound with various biological activities [23], and it is often used in medium as growth-promoting supplement that improve shoot production, SE, and root formation. Phloroglucinol can also control hyperhydricity by acting as a precursor in the lignin biosynthesis pathway and improve multiplication rate of hard-to-propagate species like woody species [24]. So far, there has been no research on the use of PG as a medium additive for coconut, but its application has been demonstrated in date palm. The addition of 396.5 µM PG to the culture medium increased date palm callus growth, shoot regeneration from callus, while reducing browning and hyperhydricity [25]. Phloroglucinol (475.8 µM) was also found to stimulate somatic embryo germination with low rates of callus formation in papaya (*Carica papaya* L.) [26]. Experimentally, TDZ has been the common cytokinin used to produce coconut clones via organogenesis [3,10,11], but PG has both auxin-like and cytokinin-like properties [24], similar to TDZ. Thus, PG may work similarly to TDZ to enhance immature inflorescence organogenesis of coconut.

Glutamine acts as a major amino donor in the synthesis of amino acids, nucleotides, and other nitrogen-containing compounds, and it is an important compound in plant nutrition and metabolism. It is also a signalling molecule regulating gene expression, plant growth, and various stress responses [27]. Up to now, there has been no research on the use of glutamine to improve direct shoot organogenesis of coconut, but the incorporation of glutamine has improved growth and vigour of pineapple (*Ananas comosus* L. Merr.) shoots. Its addition indirectly increased the endogenous levels of indole-3-acetic acid (IAA; an auxin) and isopentenyladenine (iP; a cytokinin) and improved shoot induction [28]. Several studies on shoot tip and immature inflorescence organogenesis of date palm also included glutamine in the shoot induction (0.7 to 1.4 mM) and multiplication (1.4 to 6.8 mM) media [29,30,31]. In these studies, the glutamine concentration added to the shoot multiplication medium was much higher than the amount supplemented to the shoot induction medium.

Since direct organogenesis requires significant improvement, the current study focused on establishing direct shoot organogenesis for coconut, primarily using immature inflorescence explants at the shoot induction and proliferation stages (Figure 1). Shoot tip was used in the first experiment, but inflorescence explants were used in subsequent experiments because they were immediately available in large quantities and has a lower risk of somaclonal variation. The study also investigated protocols (of different PGRs combination) suitable for inducing and proliferating shoots and aimed to improve direct shoot induction using media additives including adenine sulphate, phloroglucinol, and L-glutamine. Importantly, a histological analysis was undertaken to better understand the anatomical differences in the shoot meristems obtained from zygotic embryos and those of the shoot-like structures developed during organogenesis.

## 2. Results

### 2.1. Cytokinins Impact on Shoot Tip and Immature Inflorescence Organogenesis (Experiment 1)

As shown in the flowchart (Figure 1), the first experiment was undertaken to determine the effects of two cytokinins on two explants (shoot tips and immature inflorescence rachilla sections). Shoot tips cultured in both cytokinins (BAP and TDZ) and the control only formed single shoots (Figure 2A) after six weeks of culture. At this time, the shoot tips had produced at least one to two green leaves under the low-light conditions used. Meanwhile, inflorescence rachilla sections in all treatments enlarged in size, but they did not form shoots after four weeks of culture (Figure 2B). However, on closer examination, enlarged structures were observed at the location of the floral primordia (Figure 2C).

The effect of cytokinins and their concentrations (BAP or TDZ) were similar among the control and various treatments in terms of the percentage of explants forming a single shoot from the shoot tip explants (Figure 3A) and the percentage of inflorescence rachilla sections that remained healthy (Figure 3B). The percentage of explants that produced a single shoot at week 6 ranged from 82.5 ± 11.8% (200 µM BAP) to 100.0 ± 0.0% (control; 20 and 400 µM BAP). The percentage of inflorescence rachilla sections that survived ranged from 80.0 ± 20.0% (100 µM BAP) to 100.0 ± 0.0% (200 µM TDZ).

### 2.2. Effects of Exogenous PGRs Based on Protocols on Organogenesis from Inflorescence Sections (Experiment 2)

Several protocols previously developed for date and oil palm, as well as those developed from preliminary studies on coconut, were used in this study to investigate their effects on shoot induction and proliferation of coconut inflorescence rachilla sections. Most inflorescence sections produced multiple shoot-like structures after 16 weeks (Figure 4A–D), except in T2 (AC-free shoot induction medium supplemented with 9.8 µM 2iP), from which the explants underwent browning and eventually necrosed, presumably because this treatment was not supplemented with AC. In most other treatments with AC, the shoot like-structures continued to grow after treatment when cultured onto a PGR-free medium (Figure 4E), but they became hyperhydrated and eventually died from necrosis after two further subcultures up to 24 weeks (Figure 4F).

Significant differences were observed in the shoot induction percentage among treatments (Figure 5A). Shoot induction percentages for T1 (65.0 ± 12.6%), T3 (75.0 ± 12.6%), T5 (62.5 ± 8.5%), and T7 (80.0 ± 12.2%) were the highest and not significantly different from each other. However, these mean values were significantly higher than the shoot induction percentages of T2 (0.0 ± 0.0%) and T4 (5.0 ± 5.0%). While the shoot induction percentage of T6 (35.0 ± 20.2%) was not significantly different from all other treatments.

**Table 1 plants-14-03123-t001:** The shoot induction and multiplication media and culture conditions used in seven treatments to obtain direct shoot formation on inflorescence rachilla sections of coconut three to nine months before inflorescence opening.

Treatment	Medium and Culture Conditions	Modified from
T1	**SIM**: 100 µM TDZ; **AC**: 2.5 g L^−1^ **CP**: 12 weeks; **SF**: 4 weeks; **L**: Dark conditions	Experiment 1
	**SMM**: 100 µM TDZ; **AC**: 2.5 g L^−1^ **CP**: 4 weeks; **L**: 40 µmol m^−2^ s^−1^; **P**: 16/8 h L/D	
T2	**SIM**: 9.8 µM 2iP; **AC**: not added **CP**: 16 weeks; **SF**: 4 weeks; **L**: Dark conditions	Romyanon, Mosaleeyanon and Kirdmanee [16]
T3	**SIM**: 2.7 µM NAA + 8.9 µM BAP + 4.9 µM 2iP; **AC**: 2 g L^−1^ **CP**: 16 weeks; **SF**: 4 weeks; **L**: Dark conditions	Loutfi and Chlyah [32]
T4	**SIM**: 45.2 µM 2,4-D + 7.6 µM ABA + 14.8 µM 2iP; **AC**: 1 g L^−1^ **CP**: 16 weeks; **SF**: 4 weeks; **L**: Dark conditions	Zayed [33]
T5	**SIM**: 10.7 µM NAA + 17.8 µM BAP; **AC**: 2 g L^−1^ **CP**: 8 weeks; **SF**: 4 weeks; **L**: Dark conditions	Khierallah, Bader and Al-Khafaji [31]
	**SMM**: 14.8 µM 2iP + 8.9 µM BAP; **AC**: 2 g L^−1^ **CP**: 8 weeks; **SF**: 4 weeks; **L**: 40 µmol m^−2^ s^−1^; **P**: 16/8 h L/D	
T6	**Pre-conditioned**: 41.4 µM PIC + 4.5 µM TDZ; **AC**: 1 g L^−1^ **CP**: 4 weeks; **L**: Dark conditions	Raju [10]
	**Maintenance**: 20.7 µM PIC + 4.5 µM TDZ; **AC**: 1 g L^−1^ **CP**: 4 weeks; **L**: Dark conditions	
	**SIM**: 8.3 µM PIC + 4.5 µM TDZ; **AC**: 1 g L^−1^ **CP**: 8 weeks; **SF**: 4 weeks; **L**: 40 µmol m^−2^ s^−1^; **P**: 16/8 h L/D	
T7	**Pre-conditioned**: 22.6 µM 2,4-D + 14.8 µM 2iP; **AC**: 1.5 g L^−1^ **CP**: 4 weeks; **L**: Dark conditions	Gadalla [34]
	**Maintenance:** 4.5 µM 2,4-D + 14.8 µM 2iP + 5.4 µM NAA + 11.4 µM IAA; AC: 1 g L^−1^ **CP**: 8 weeks; **SF:** 4 weeks; **L:** Dark conditions	
	**SIM**: 2.3 µM 2,4-D + 14.8 µM 2iP + 2.7 µM NAA; **AC**: 0.5 g L^−1^ **CP**: 4 weeks; **L**: Dark conditions	

Key: AC: activated charcoal, CP: culture period, L: light intensity, L/D: light/dark, P: photoperiod, SF: subculture frequency, SIM: shoot induction medium, SMM: shoot multiplication medium, and T: treatment.

A significant difference was observed between treatments for the number of shoot-like structures produced per explant (Figure 5B). Treatment 7 (7.7 ± 2.2) recorded the highest number of shoot-like structures per explant, and it was significantly higher than the values from T2 (0.0 ± 0.0) and T4 (0.6 ± 0.6). The mean number of shoot-like structures per explant for T1 (5.9 ± 1.2), T3 (5.6 ± 0.6), and T5 (5.5 ± 1.0) were not significantly different from each other, and these values were comparable to those from T4, T6 (3.7 ± 1.1), and T7. As a result, T1, T3, T5, and T7 were found suitable for shoot induction and proliferation of coconut inflorescence rachilla sections. However, T3, which recorded a high shoot induction percentage (75%) and a comparatively higher number of shoot-like structures per explant (with a lower standard error of 5.6 ± 0.6), was the frontrunner; hence, the PGR combination used in this treatment was then applied to the subsequent experiments (Experiments 3 to 5).

### 2.3. Influence of AdS on Shoot Induction from Immature Inflorescence Sections (Experiment 3)

After 12 weeks of culture, all inflorescence rachilla sections treated with or without AdS increased in size and formed distended globular-like structures at the location of the floral primordia (Figure 6), with hyperhydricity being observed to have occurred in some of these structures. There were no apparent differences in morphology among the responses of the explants treated with or without AdS. The distended globular-like structures were whitish in colour, as this stage of shoot induction had been conducted under dark conditions.

Based on the results, the supplementation of various concentrations of AdS into the Y3 culture medium resulted in a significant effect on the shoot induction percentage (Figure 7). The highest shoot induction percentage (50 ± 10%) was obtained in explants cultured in medium containing with 217 µM AdS, and this value was significantly higher than the control without AdS (ca. 5%). The rachilla sections treated with 326 µM AdS (45 ± 5%) had a comparable shoot induction percentage with those treated with 217, 434 (ca. 10%), and 543 µM AdS (ca. 30%). When the concentration of AdS was higher than 217 µM, a lower shoot induction percentage was observed.

### 2.4. PG for Shoot Induction from Immature Inflorescence Sections (Experiment 4)

In this experiment, the explants cultured on a medium supplemented with various concentrations of PG increased in size and took on a whitish globular appearance (Figure 8). Some structures became hyperhydrated with a translucent appearance (Figure 8B). The rachilla sections on the medium supplemented with or without PG had a similar morphology.

The addition of PG into the shoot induction medium had no significant impact on the percentage of rachilla sections with enlarged structures (Figure 9). The shoot induction percentage ranged from 4.4 ± 4.4% to 20.0 ± 6.7%. As the concentration of PG increased, an increasing trend in the shoot induction percentage was observed. However, the PG-treated rachilla sections had a comparatively lower percentage of shoots induced than the AdS-treated explants. Therefore, it is suggested to add AdS during shoot induction from inflorescence explants of coconut than without as in the control.

### 2.5. L-Glutamine Effect on Shoot Induction from Immature Inflorescence Sections (Experiment 5)

The explants incubated on a Y3 medium supplemented with or without L-glutamine were not significantly different in terms of their morphology (Figure 10). All explants enlarged in size and produced whitish globular structures at the site of the floral primordia. Several larger globular structures that were in direct contact with the medium became translucent due to hyperhydricity.

The addition of L-glutamine (Figure 11) had no significant impact on the shoot induction percentage formed on the inflorescence rachilla sections. The shoot induction percentage ranged from 80.0 ± 10.0% to 100.0 ± 0.0%. The explants treated with L-glutamine generally had a higher shoot induction percentage than those cultured without L-glutamine. Also, an increasing trend was observed in the shoot induction percentage as the concentration of L-glutamine supplemented to the medium increased. A 100% shoot induction percentage was recorded on a medium supplemented with 3400 µM L-glutamine, but the shoot induction percentage dropped (93.3 ± 3.3%) when the concentration of L-glutamine was doubled to 6900 µM.

### 2.6. Histological Analysis

The histological comparison of the meristematic regions of the Sampoorna zygotic embryos (0 to 4 weeks old) (Figure 12A–C) and those of the shoot-like structures on inflorescence rachilla sections (as seen in Experiment 2; Figure 12D) was used to identify the nature of the shoot-like structures.

In the case of germinated zygotic embryos, there was a noticeable size increase in the meristematic zone after four weeks of incubation. At this stage, more leaf primordia were observed surrounding the shoot apical dome, which was rich in meristematic cells (densely stained in blue by naphthol blue black). In the case of the meristematic zones observed in 0- to 4-week-old zygotic embryos, they were surrounded by two leaf primordia with high meristematic activity. In addition, numerous vascular bundles were observed, consisting of both short and long strands running among the young and mature leaf primordia.

The shoot-like structures (Experiment 2) developed on inflorescence explants had a similar structure to the shoot meristematic regions observed on the zygotic embryos (Figure 12D). A small domed area, rich in meristematic cells and with several vascular tissues, were observed in the surrounding leaf primordia-like tissues. The enlarged shoot-like structures produced by the floral primordia treated with cytokinins and auxins resembled true shoot-like structures (Figure 12D). However, they still did not develop further into true shoots after being sub-cultured onto a Y3 PGR-free medium due to hyperhydricity and necrosis in all treatments.

A further series of inflorescence rachilla sections at different developmental stages were selected to observe differences in terms of their meristematic region size, their locations, and frequency. This allows determination of a suitable maturity stage for immature inflorescence organogenesis to be used for Experiments 3 to 5. There were numerous meristematic floral primordia (enlarged picture in Figure 13) along the whole inflorescence rachilla of −2 developmental stage (Figure 13). In addition, there were countless vascular tissues or bundles being observed as long, discrete strands in the middle of the longitudinal section of the inflorescence explant. The meristematic areas were characterized by the presence of numerous meristematic cells with blue-stained nuclei and arranged at the periphery of the meristematic dome.

Similar structures were observed in the inflorescence rachillas from −3, −4, and −5 stages (Figure 14). However, the meristematic areas became smaller in size and fewer in number as the developmental stage of inflorescence decreased from −3 to −5. There were less vascular tissues in the middle of the longitudinal sections as the explants became more immature. The floral primordia at the tip of the inflorescence rachilla were observed to be smaller than those at the basal area. Although the floral primordia of inflorescence rachilla from −2 stage were higher in frequency and bigger in size, these explants were more mature and the floral primordia were not likely to revert into vegetative forms when compared to −3, −4, and −5 developmental stages.

## 3. Discussion

Although SE offers a promising method for the clonal propagation of coconut, it is still important to understand the potential of the less-studied micropropagation pathway of direct shoot organogenesis (Figure 1). This method is considered beneficial in terms of its lower risk of causing somaclonal variation [35] and may be a faster way to produce plantlets [16]. There are four stages to undertake direct shoot organogenesis, viz. (1) shoot bud induction, (2) shoot bud multiplication, (3) shoot bud elongation, followed by (4) root formation on the developed shoots [16]. Of these stages, the first stage is the most critical [35].

Cytokinins such as BAP and thidiazuron (TDZ) have been found to induce direct shoot formation from coconut vegetative shoots or inflorescence explants [9,10]. Therefore, the effect of various concentrations of BAP and TDZ on the formation of shoots from meristem and immature inflorescence explants was determined in this study. For shoot tip explants, it was found that only a single shoot (Figure 2) could be formed regardless of the type and concentration of cytokinins used (Figure 3). This was likely because mechanical injury at the growing point may be essential for multiple shoot formation [3,6], and in this present study, mechanical injury was not applied. On the other hand, the inflorescence explants used had enlarged in size and formed large structures at the site of each floral primordium. Thidiazuron was also found to be the best cytokinin to support this growth from inflorescence explants, based on the survival of inflorescence sections (Figure 3). Subsequent experiments used inflorescences as explants due to the initial positive results, because they were somatic tissues that ensure true-to-type plants during micropropagation, and because they were much more numerous than seedling meristems.

The results indicated a strong potential for direct shoot organogenesis using immature inflorescence explants as an alternative micropropagation pathway for coconut. To promote inflorescence organogenesis, TDZ was found to be a suitable PGR and was effective even when applied alone (Figure 5). This may be because TDZ has both auxin- and cytokinin-like activity [36]. Although other treatments consisting of several PGR combinations gave comparable results to TDZ, it will be easier to use TDZ in all future research on shoot regeneration via direct organogenesis for coconut. In all treatments tested (Experiment 2) except for the treatment without AC, numerous shoot-like structures (Figure 4) that resemble true shoots when histologically compared with the shoot meristematic regions of zygotic embryos were formed (Figure 12). When the immature inflorescence rachilla explants were cultured in a medium without AC, this led to tissue browning and eventual death (Figure 5). However, in an earlier study using vegetative meristems in the presence of TDZ (1 µM [3]), the presence of AC in the medium was not necessary to protect them from oxidation reactions. Therefore, an alternative to AC may be needed to examine the effects of PGRs on shoot induction from inflorescence explants in the future. Antioxidants such as ascorbic acid and citric acid could be alternatives, as they have been found to significantly reduce browning of coconut inflorescence and meristem explants [37]. In addition, 2-aminoindan-2-phosphonic acid is another possible option that has demonstrated a positive effect in overcoming browning during the formation of coconut EC [38].

The developmental stage of the immature inflorescence rachilla sections used is known to influence shoot induction in coconut palm [10]. Although mature inflorescence (−2 stage; Figure 13) carried more floral meristems and a bigger meristematic area where shoots may develop, they were more likely to produce floral structures instead of shoots, presumably due to their advanced maturity. Hence, the immature inflorescence rachilla sections taken from the −3, −4, and −5 stages of growth (3, 4, and 5 months before spathe opening) were shown to be more suitable for direct shoot organogenesis based on histological analysis (Figure 14). This view is supported by Raju [10], who discovered that coconut shoot induction occurred via organogenesis in spadices of 10 to 12 cm inner spathe length, whereas callus or floral formation was observed when spadices of less than 10 cm or more than 13 cm inner spathe length were used, respectively. Hence, inflorescence rachilla sections of −4 (ca. 11 to 13 cm inner spathe length) or −5 (ca. 7 to 9 cm inner spathe length) developmental stages were used for Experiments 3 to 5.

Several additives including AdS, PG, and glutamine were incorporated to the culture medium to improve direct shoot induction from immature inflorescences. The additive phloroglucinol was found not to be useful in improving the shoot induction percentage (Figure 9). Phloroglucinol produced a negative impact on the explants, likely due to the nature of PG acting as a phenol, causing tissue browning, while AdS and L-glutamine produced more positive impacts (Figure 6 and Figure 10). The study demonstrated that supplementation of AdS to the medium significantly improved the percentage of shoot-like structures being formed (Figure 7). This was likely due to the synergistic effect of AdS, which can reinforce the influence of other PGRs in the medium, as shown in the study of Naaz, Shahzad, and Anis [21]. The addition of L-glutamine also produced a higher shoot induction percentage, ca. 83 to 100% (Figure 11), this was likely due to its function in nitrogen metabolism [27] and its positive influence on the endogenous levels of auxins and cytokinins [28] during shoot induction.

A possible genotype influence was observed in the Experiments 3 to 5 undertaken on the CAIRNS variety. The control treatments gave variable shoot induction percentages [viz. Experiments 3 (5%), 4 (11%), 5 (80%)] when the immature inflorescence explants of different palms were used as explants. This is similar to the observations made of the genotypic influence on embryo germination and callus induction percentages discovered among the different varieties used in the SE pathway [39]. Hence, variation in shoot-like structure production depended on both the genotype studied and the PGRs used. It will be important to tailor approaches for individual varieties of coconut to enable organogenesis to be developed as a successful micropropagation pathway.

It seems that direct shoot organogenesis from immature inflorescence explants has the potential to clonally propagate coconut; however, it is not yet superior to SE in a commercial setting. This is because the shoot-like structures produced in this study have not been able to grow further into plantlets due to hyperhydricity and necrosis. Hence, a substantial amount of further research is still required to further reduce the hyperhydricity (e.g., changing gelling agent, the nutrients supplied, the medium water potential, the culture and vessel environment) and necrosis, to better understand the process (including root formation), increase the number of clones, and ensure the plantlets formed are true-to-type clones.

## 4. Materials and Methods

### 4.1. Plant Materials

Shoot tips from in vitro seedlings and immature inflorescences were the explants used in Experiment 1, while for all other experiments only inflorescence explants were used. To produce in vitro seedlings, zygotic embryos were isolated from 12-month-old Nias Yellow Dwarf (NYD) fruit obtained from Pangandaran, West Java, Indonesia and cultured, with monthly subculture according to the procedures of Sisunandar et al. [40], until they produced 3-month-old in vitro-growing seedlings (Figure 15A). The work was undertaken in the laboratory of Professor Sisunandar by Dr Eveline Kong while on placement at the Coconut Research Centre, the University of Muhammadiyah Purwokerto, Indonesia. Shoot tips were isolated from the 3-month-old seedlings (2 to 3 cm long) in a laminar air flow (LAF) hood with the aid of a dissecting microscope (Leica EZ4). First, the larger leaves, and roots (if any) were removed, followed by thin slicing of leaves in a sterile liquid Eeuwens Y3 medium [18] containing ascorbic acid (200 mg L^−1^) until the last one or two leaf primordia surrounding the meristem were exposed (Figure 15B). Then, the shoot tips containing meristems (each ca. 2 to 4 mm^2^) were carefully excised and cultured onto a medium as described for Experiment 1 (below). A video of shoot tip extraction can be found in Appendix A.

To obtain inflorescence explants, intact inflorescences were collected from a known Indonesian (Banyumas Tall; BT—used in Experiment 1) and an unknown Australian coconut variety (referred to as CAIRNS—used in Experiments 2 to 5). Immature inflorescences from BT palms (8-years old) were obtained from Banyumas, Central Java, Indonesia and delivered to the University of Muhammadiyah Purwokerto, while Cairns palms (9- to 20-years old) collected from Cairns, North Queensland were transported to the University of Queensland (UQ) Gatton Campus. In both cases, immature inflorescences of various developmental stages were collected and randomly allocated into treatments. The palm leaf bases were first removed using a small chainsaw, then the inflorescences were extracted from the axil of each leaf using a clean scalpel blade. A video of inflorescence extraction from a CAIRNS coconut crown can be found in Appendix A.

Prior to experimentation, the inflorescences of both BT and CAIRNS varieties (with different developmental stages ranging from −2 to −9, with ages between two and nine months before opening), still enclosed in the double-layered spathe (Figure 16A), were surface sterilized. Initially, with a 70% (*v*/*v*) ethanol solution (UQ Science Store, Brisbane, Australia; US004291) for 1 min followed by a 6% (*w*/*v*) pool chlorine solution (prepared from the commercial brand (HY-CLOR, Sydney, Australia; HYCG02), which contained 650 g kg^−1^ calcium hypochlorite) for 30 min with gentle shaking in a LAF hood. Then, the explants were rinsed with sterile liquid Y3 medium consisting of Y3 nutrients, sucrose (ChemSupply, Adelaide, South Australia, Australia; SA030; 30 g L^−1^), and ascorbic acid (Sigma, Melbourne, Victoria, Australia; A-7506; 200 mg L^−1^) for 10 min. The outer spathe was then removed, and the inner tissues were surface sterilized with 6% (*w*/*v*) pool chlorine solution for 10 min with gentle shaking, followed by a double rinsing with sterile liquid Y3 medium for 10 min. The inner spathe was then removed, and the rachillas (different sections used in different experiments; Figure 16B) were detached under sterile liquid Y3 medium. The tip and basal sections of each rachilla were cut off and the mid sections were cut into ca. 2 to 4 mm long explants. Each rachilla mid-section represented an explant and cultured onto a medium as described in each experiment.

#### Plant Materials for Histological Analysis

There were three types of plant materials selected for histological studies: the meristematic regions of zygotic embryos, the shoot-like structures developed during immature inflorescences organogenesis, and explants taken from various developmental stages of immature inflorescences.

The zygotic embryos of Sampoorna variety (a hybrid cross between Indian Yellow Dwarf and Andaman Tall) were obtained from India through in-country collection and preparation, shipment, and quarantine passage. Upon arrival, the embryos were removed from the culture tubes and surface sterilized. After surface sterilization, the embryos were cultured onto an embryo germination medium as previously described [39] for two to four weeks under dark conditions. The embryos were then used for histological studies either directly (not surface-sterilized) or after 2- or 4-weeks of germination. As for the shoot-like structures developed from CAIRNS inflorescence explants, they were obtained after a further four weeks of culture at the end of Experiment 2, as described later (Section 4.3). These explants were used directly for histological studies after removing the surrounding culture medium.

The immature inflorescence explants of the aromatic Dua Dua (AR) from Vietnam were first processed in Vietnam. This involved the surface sterilizing the double-layered spathe enclosing the rachillas with 70% (*v*/*v*) ethanol for 1 min, followed by 0.5% (*w*/*v*) sodium hypochlorite for 15 min, and rinsing with sterile distilled water three times in the LAF hood. After removing the outer spathe, the inner tissues were surface sterilized with 0.5% (*w*/*v*) sodium hypochlorite for 2 min, followed by a triple rinse with sterile distilled water. The rachillas, still enclosed within the inner spathe were inoculated onto a semi-solid Y3 medium supplemented with 2.5 g L^−1^ phytagel, 2.5 g L^−1^ AC, and 30 g L^−1^ sucrose and placed into 25 mL Eppendorf tubes. They were then shipped to UQ under the import permit, and upon arrival, they were surface sterilized with 70% (*v*/*v*) ethanol for 1 min followed by 6% (*w*/*v*) pool chlorine solution for 10 min and rinsed twice with sterile liquid Y3 medium (10 min) in the LAF hood. The inner spathe was removed and the rachillas were detached and used directly for histological analysis according to the protocol described later (Section 4.5). An experimental flowchart for the present study is shown below (Figure 17).

### 4.2. Cytokinins Impact on Shoot Tip and Immature Inflorescence Organogenesis (Experiment 1)

This experiment was conducted at the Coconut Research Centre, The University of Muhammadiyah Purwokerto, Indonesia. The isolated shoot tips from 3-month-old seedlings of NYD and the extracted rachillas at the −3 to −6 stages of development (ca. 3 to 6 months before opening) from two palms of BT were used as the explants in this experiment. Both forms of explants were cultured onto a shoot induction medium (SIM) which consisted of Y3 medium, AC (Sigma, C9157; 2.5 g L^−1^), sucrose (30 g L^−1^), Gelzan (Sigma, G1910; 3 g L^−1^), and various concentrations of BAP (Sigma, B3408; 0, 20, 40, 100, 200, or 400 µM) or TDZ (Sigma, P6186; 0, 20, 40, 100, or 200 µM) in Petri dishes (LabServ^®^, Auckland, New Zealand; LBS60014X, 90 × 25 mm diameter/height; 42 mL media). A PGR-free Y3 medium was used as the control. All cultures were incubated under a 16/8 h (light/dark) photoperiod at a low white light intensity of 5 µmol m^−2^ s^−1^ for four to six weeks. The temperature was maintained at 27 ± 2 °C throughout the culture period. The number of explants that produced at least a single shoot from the shoot tips at week 6, and the number of immature inflorescence explants that survived at week 4, were recorded. The immature inflorescence explants that showed growth enlargement and with no serious browning, were counted as surviving explants.

A single shoot tip isolated from an in vitro seedling or a section of rachilla was taken as one explant. Each treatment began with five replicates of 10 to 21 explants per replicate, but due to contamination, treatments had between two to five replicates, while the total number of explants per treatment ranged from 41 to 71 explants. A randomized complete block design was used, with the experiment blocked by time and plant material, as the shoot tips were isolated at different times and the inflorescence explants were isolated from two palms. The data collected were then analyzed using a two-way analysis of variance (ANOVA) using Minitab 17 Statistical Software.

### 4.3. Effects of Exogenous PGRs Based on Protocols on Organogenesis from Inflorescence Sections (Experiment 2)

From Experiment 2 onwards, only immature inflorescences were used as such explants revealed more tissue enlargement (propensity to form shoots) than shoot tips in Experiment 1, which could only produce one shoot per explant.

Rachillas from the −3 to −9 stages of development were isolated from a single CAIRNS palm and used as explants. The explants were cultured onto several media [viz. a pre-conditioning, a maintenance, a shoot induction media (SIM), and a shoot multiplication media (SMM)], whereby each medium consisted of Y3 medium, sucrose (30 g L^−1^), Gelzan (3 g L^−1^), AC, and PGRs, except for one (T2), which used a 0.5 Y3 medium. Information on the concentration of AC for each culture medium and the concentration of PGRs [viz. TDZ, 2iP (Sigma, D7660), 1-naphthaleneacetic acid (NAA; Sigma, N0640), BAP, 2,4-D (Sigma, D7299), abscisic acid (ABA; Sigma, A1049), PIC (Sigma, P5575), IAA (Sigma, I2886), and KIN (Sigma, K3253)] used in each protocol are shown in Table 1 (See Section 2.2). All cultures were contained within Petri dishes and incubated under culture conditions, with subculture frequencies as shown (Table 1; see Section 2.2) and at a temperature of 27 ± 1 °C. Then, the shoot induction percentage and the number of shoot-like structures per explant were determined after 16 weeks. The shoot-like structures that formed were selected and further cultured onto a PGR-free Y3 medium contained in 250 mL glass bottles (Austratec Pty. Ltd., Victoria, Australia; C926), supplemented with sucrose (30 g L^−1^), Gelzan (3 g L^−1^), and AC (2.5 g L^−1^) under illuminated conditions with a photoperiod of 16/8 h (light/dark) at 40 µmol m^−2^ s^−1^ and a temperature of 27 ± 1 °C for another eight weeks.

The shoot induction percentage was based on four replicates per treatment (T) with 10 explants in each replicate. Each rachilla section represented one explant. Documenting the number of shoot-like structures per explants was based on the number of explants with shoots induced only, so there were four replicates per treatment with an average of four to eight explants per replicate. The experimental design was a completely randomized design (CRD) and all data collected were analyzed using a one-way ANOVA.

### 4.4. Effect of AdS (Experiment 3), PG (Experiment 4) and L-Glutamine (Experiment 5) on Shoot Induction from Immature Inflorescence Sections

For Experiment 3, rachillas from −5 stage of development were isolated from two CAIRNS palms, while the rachillas from two CAIRNS palms at the −4 stage of development were used for Experiments 4 and 5, with one palm for each experiment. The rachillas were excised into sections of 2 to 4 mm long as previously described (Section 4.1). The explants were cultured onto a SIM which consisted of Y3 medium, sucrose (30 g L^−1^), Gelzan (3 g L^−1^), AC (2.5 g L^−1^), NAA (2.7 µM), BAP (8.9 µM), and 2iP (4.9 µM).

In Experiment 3, SIM was additionally supplemented with various concentrations of AdS (Austratec Pty. Ltd., A545; 0, 109, 217, 326, 434, and 543 µM). Adenine sulphate was added to the medium before autoclaving. As for Experiments 4 and 5, different concentrations of PG (PhytoTech Labs, Shawnee Mission, KS, USA; P694; 0, 198, 397, 793, 1189, and 1586 µM) or L-glutamine (Sigma, G8540; 0, 700, 1400, 2100, 3400 and 6900 µM) were additionally supplemented to the SIM, respectively. All explants were cultured in Petri dishes, and incubated under total darkness at a temperature of 27 ± 1 °C. The cultures were sub-cultured every 4 weeks for a duration of 12 weeks. The number of explants with enlarged structures was recorded and the shoot induction percentage was calculated.

Each rachilla section represented one explant and there were three replicates per treatment. Experiments 3 and 5 had 10 explants in each replicate, whereas Experiment 4 had 15 explants per replicate. A CRD was used, and the shoot induction percentage was calculated and analyzed using a one-way ANOVA.

### 4.5. Histological Analysis

Germinating zygotic embryos from the Indian variety, Sampoorna (at either 0, 2, or 4 weeks after culture), explants with shoot-like structures obtained from CAIRNS inflorescence explants in Experiment 2 (5-month-old), and immature inflorescence explants of AR from Vietnam (at −2, −3, −4, and −5 developmental stages), were selected for histological study. This study on zygotic embryos was used to determine the basic structure and growth pattern of embryo plumules for comparison with the shoot-like structures developed from the CAIRNS variety. The study on the different developmental stages of immature inflorescence explants was conducted to determine a suitable explant age for immature inflorescence organogenesis for Experiments 3 to 5.

The histological studies were undertaken using protocols recommended by the UQ School of Biomedical Science Histology Facility. Before transport to the Histology Facility, the samples were fixed in a formalin–acid–alcohol fixative solution consisting of 50% ethanol, 10% formaldehyde, and glacial acetic acid in an 18:1:1 ratio for 24 h, then washed under running distilled water for 15 min. The samples were then placed into cassettes (Macrosette^®^, Redding, CA, USA; 27166) and submerged in 70% (*v*/*v*) ethanol before being transported to the facility for processing. The histological processing methods used were the same as previously described [39], apart from the Sampoorna embryos (containing shoot meristems), which required additional preparation. These embryos were dehydrated in a more extended series of ethanol solutions of increasing concentration [70, 90, 95, and three times in 100% (*v*/*v*)] before being covered twice with xylene for 10 min in a vacuum chamber (0.4 MPa) at room temperature. They were then placed onto a stirrer for 50 min at standard room temperature and pressure. The xylene-treated embryos were then covered with liquid paraffin wax and placed into the vacuum chamber (0.4 MPa) for 1 h at 65 °C and this treatment was repeated. Then, the samples were covered with fresh paraffin wax and placed in the vacuum chamber (0.4 MPa) for three days at 65 °C before being mounted on an embedding station (Medite, TES Valida; Thermo Scientific, Microm EC 350-2). The wax-covered samples were sectioned into 6 µm sections using a rotary microtome (Leica Biosystems, Nussloch, Germany; RM2245) and collected on microscope slides maintained in a water bath (40 °C). The slides were placed in the oven (38 °C) overnight to remove moisture before performing the staining procedure. The staining steps from dewaxed samples in xylene until staining with 1% periodic acid were the same as previously described [39]. Following staining in 1% periodic acid, the sections were washed in distilled water (5 min) and stained with Schiff’s reagent (20 min), then covered twice with 0.05 M sulphuric acid (2 min) and finally washed in running distilled water (2 min). The slides were then covered with preheated solutions (60 °C) of 1% naphthol blue black in 7% acetic acid (2 min) and rinsed in distilled water (2 min). The stained sections were then dehydrated in a series of ethanol solutions (70, 90, and three times in 100%), each concentration for 2 min and cleared in xylene three times, 2 min for each step. The sections were then mounted with a coverslip using DePex mounting medium (Ajax, Melbourne, Victoria, Australia; AJA3197) and left to dry overnight in a fume hood. The slides were scanned into soft copy using a digital slide scanner (Leica, Aperio XT Brightfield Slide Scanner) and observed for relevant structures using the Aperio ImageScope software (version 12.3.0.5056). The photos were taken at magnifications from 7 to 65× using the slide viewing software.

## 5. Conclusions

This study presented the use of immature inflorescences as suitable explant for direct shoot organogenesis of coconut because only a single shoot was produced from shoot tip organogenesis. Hence, shoot tip organogenesis requires further studies that should focus on using an AC-free medium [3]. On the other hand, the results of the present study have shown that direct shoot organogenesis using immature inflorescence rachilla sections can produce large numbers of shoot primordia. While those shoots still need to be conveyed into healthy rooted shoots and then plantlets. Once plantlet production is achieved, the immature inflorescence organogenesis pathway could be used as an alternative to SE in the clonal propagation of coconut, as the results showed that from one single rachilla section, ca. six to eight shoots can be produced. In the present study, the addition of AdS (217 µM) and L-glutamine (3400 µM) in the presence of AC has been identified as important additives for shoot production due to the positive influences observed in immature inflorescence organogenesis. Therefore, future research on immature inflorescence organogenesis should focus on better understanding the process (including root formation) and genotype influence, employing antioxidants to reduce browning problems, and experimenting with ethylene inhibitors or adjusting culture media strength to prevent hyperhydricity in coconut inflorescence explants.

## Figures and Tables

**Figure 1 plants-14-03123-f001:**
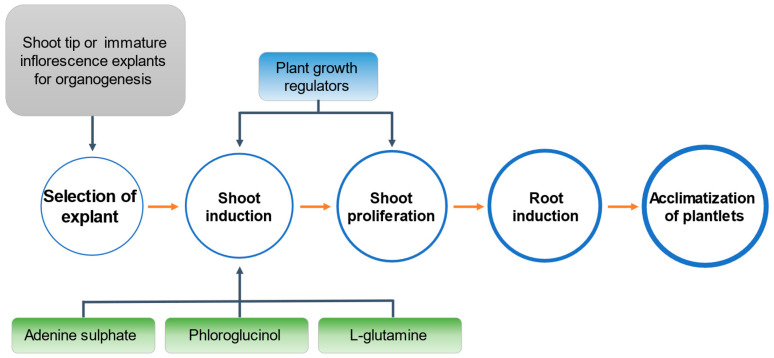
Start-to-end flowchart of direct shoot organogenesis in coconut (*Cocos nucifera* L.), showing the experiments covered in the present study for the first three stages of the process.

**Figure 2 plants-14-03123-f002:**
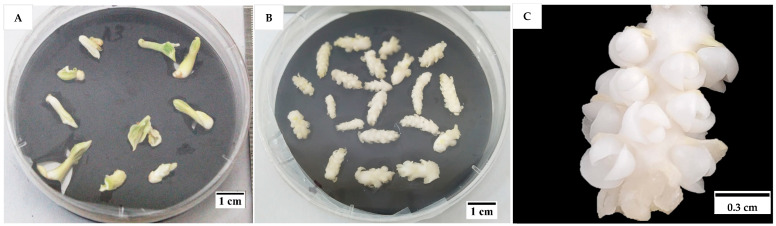
(**A**) A single shoot was formed from each shoot tip after six weeks of incubation on plant growth regulator (PGR)-free shoot induction medium. (**B**) No shoots were formed from inflorescence rachilla sections after four weeks of incubation on shoot induction medium supplemented with 100 µM thidiazuron (TDZ) in the presence of activated charcoal (AC). (**C**) An enlarged image of the structures formed from the inflorescence rachilla sections (of −4 developmental stage) cultured on medium supplemented with TDZ.

**Figure 3 plants-14-03123-f003:**
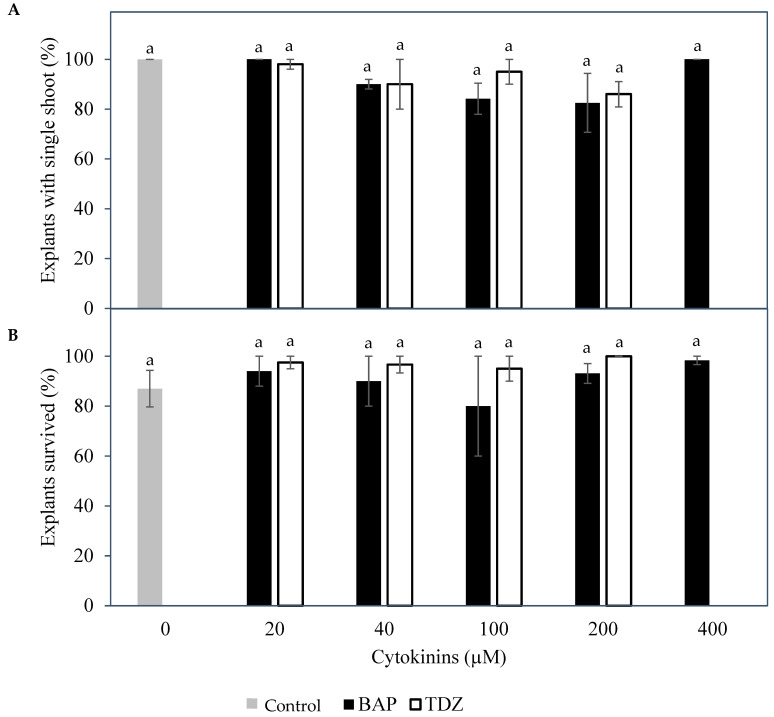
The effect of control (
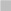
) and various concentrations of two cytokinins, 6-benzylaminopurine, BAP (
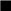
) and TDZ (
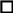
), on (**A**) the percentage of meristem explants producing a single shoot and (**B**) the percentage of inflorescence rachilla sections that survived the treatments. The mean ± standard error is shown on each bar for each cytokinin concentration, and data with the same letter were not significantly different (Tukey’s test, *p* ≤ 0.05); n = 5.

**Figure 4 plants-14-03123-f004:**
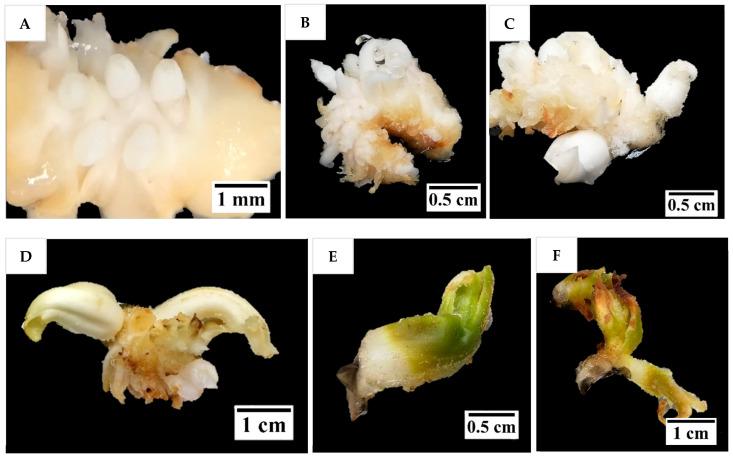
The formation of shoot-like structures on inflorescence rachilla sections after (**A**) 4, (**B**) 8, (**C**) 12, and (**D**) 16 weeks of treatment. The shoot-like structures first became (**E**) greenish after being cultured under illuminated conditions for four weeks on a PGR-free Y3 medium, but (**F**) then eventually suffered from necrosis after being sub-cultured for another four weeks.

**Figure 5 plants-14-03123-f005:**
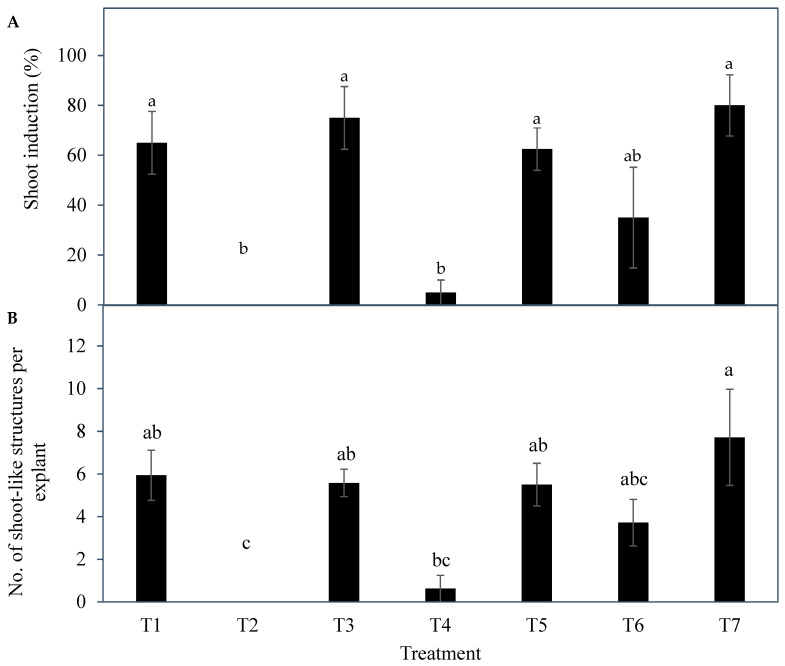
The effect of different treatments (T1 to T7; see Table 1) on (**A**) the shoot induction rate and (**B**) the number of shoot-like structures produced per explant (inflorescence rachilla section). Bars represent mean ± standard error. The mean for each treatment with the same letter is not significantly different for each parameter (Tukey’s test, *p* ≤ 0.05); n = 4. Key: No explant formed shoot-like structures in T2 due to severe browning and necrosis.

**Figure 6 plants-14-03123-f006:**
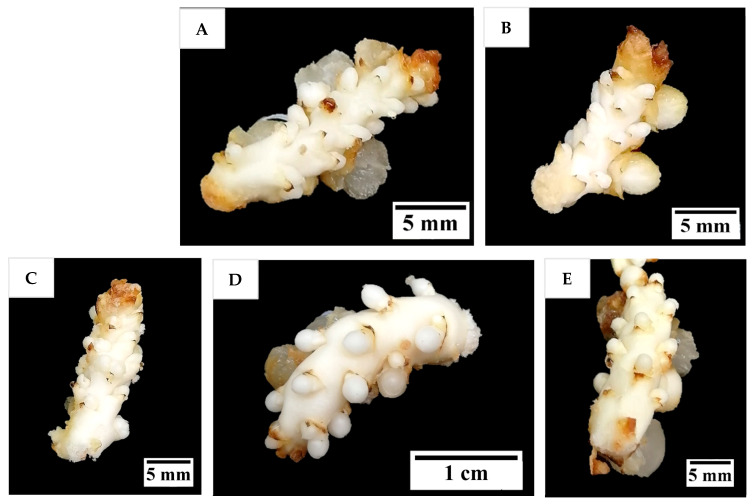
The effect of adenine sulphate on the formation of enlarged structures from inflorescence rachilla sections (of −5 developmental stages), in (**A**) 0, (**B**) 217, (**C**) 326, (**D**) 434, and (**E**) 543 µM adenine sulphate (AdS) after 12 weeks incubation under dark conditions.

**Figure 7 plants-14-03123-f007:**
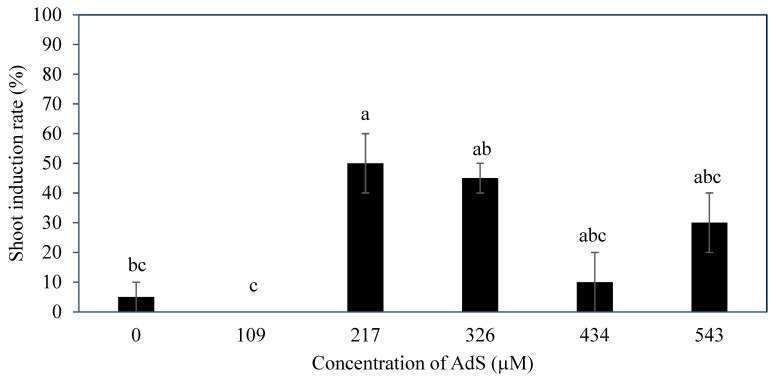
The effect of adenine sulphate concentration on the percentage of shoots induced from inflorescence rachilla sections after 12 weeks. The mean values ± standard error are shown in bars for each treatment, and treatments followed by different letters are significantly different (Tukey’s test, *p* ≤ 0.05); n = 2.

**Figure 8 plants-14-03123-f008:**
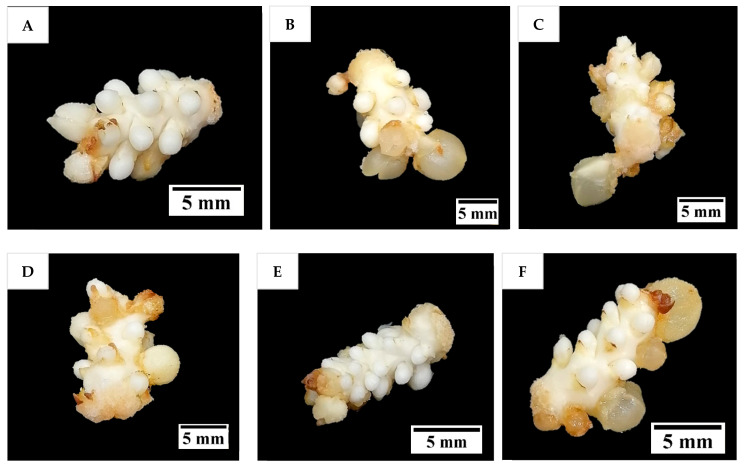
The formation of enlarged structures from the inflorescence rachilla sections (of −4 developmental stage), in (**A**) 0, (**B**) 198, (**C**) 397, (**D**) 793, (**E**) 1189, and (**F**) 1586 µM of phloroglucinol for 12 weeks incubation under dark conditions.

**Figure 9 plants-14-03123-f009:**
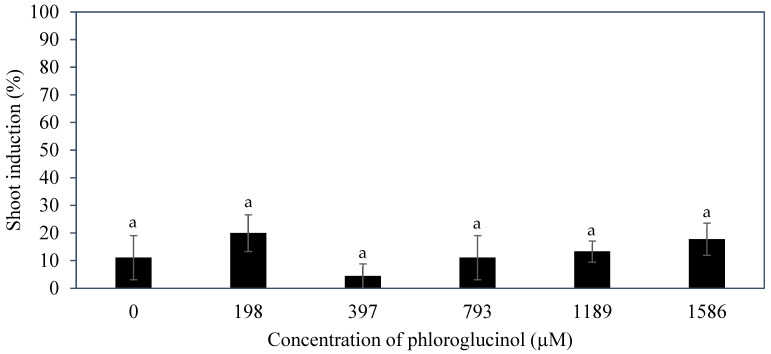
The effect of various concentrations of phloroglucinol on the percentage of shoots induced from inflorescence rachilla sections after 12 weeks. The means ± standard error for the treatments with the same letter are not significantly different (Tukey’s test, *p* ≤ 0.05); n = 3.

**Figure 10 plants-14-03123-f010:**
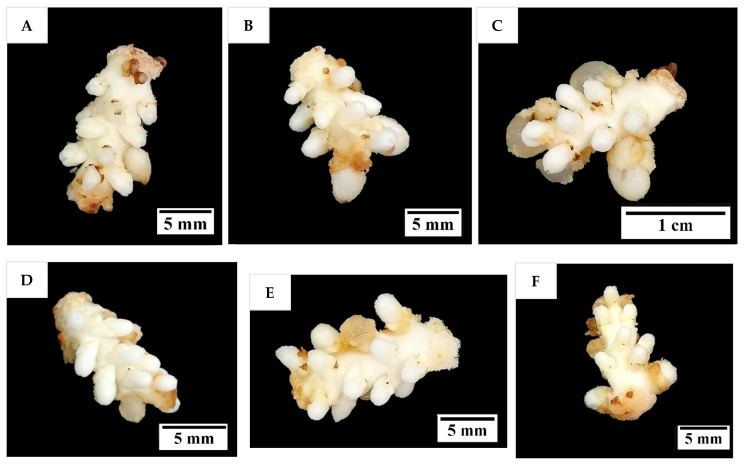
The influence of various concentrations of L-glutamine on the formation of enlarged structures from inflorescence rachilla sections (of −4 developmental stage), in (**A**) 0, (**B**) 700, (**C**) 1400, (**D**) 2100, (**E**) 3400, and (**F**) 6900 µM L-glutamine for 12 weeks incubation under dark conditions.

**Figure 11 plants-14-03123-f011:**
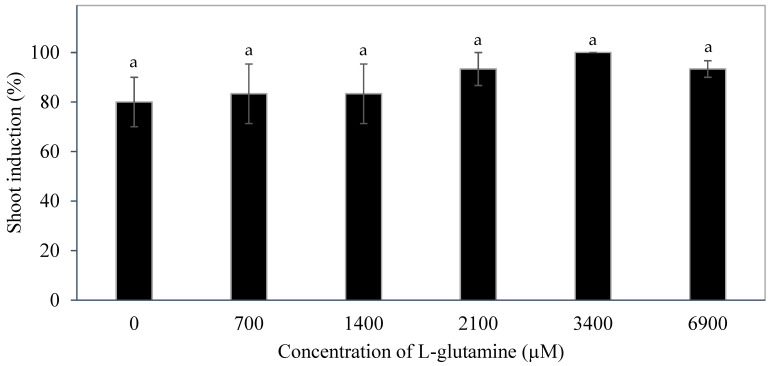
The effect of different concentrations of L-glutamine on the percentage of shoots induced from inflorescence rachilla sections after 12 weeks. The mean values ± standard error for all treatments with the same letter are not significantly different (Tukey’s test, *p* ≤ 0.05); n = 3.

**Figure 12 plants-14-03123-f012:**
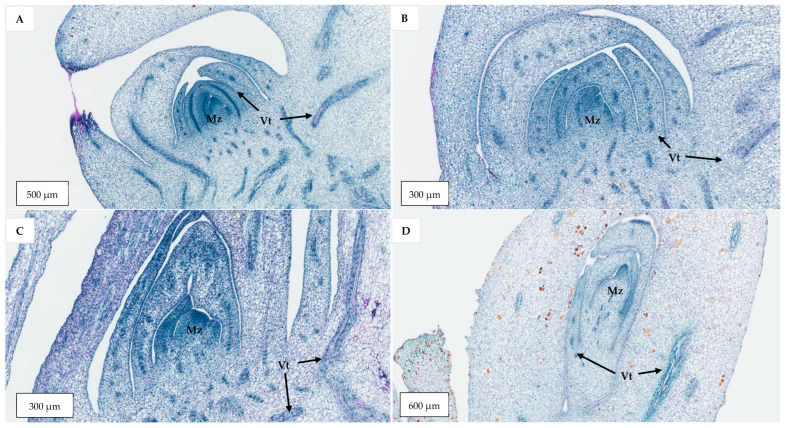
A series of longitudinal sections taken through the shoot meristem region of germinating zygotic embryos cultured on a PGR-free Y3 medium for (**A**) 0, (**B**) 2, and (**C**) 4 weeks, and (**D**) a longitudinal section taken through the shoot-like structure developed from an immature inflorescence rachilla section. The meristematic zone (Mz) was densely stained in blue and vascular tissues (Vt) were also present.

**Figure 13 plants-14-03123-f013:**
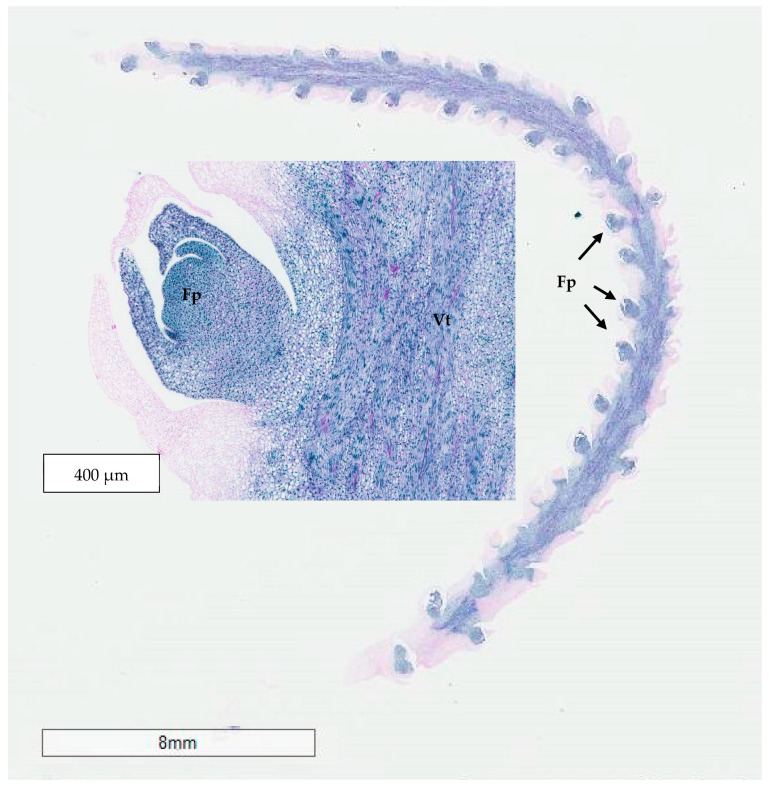
A longitudinal section taken through an inflorescence rachilla from the −2 developmental stage with an enlarged section of a floral primordium (Fp). Vascular tissues (Vt) were found in the middle section.

**Figure 14 plants-14-03123-f014:**
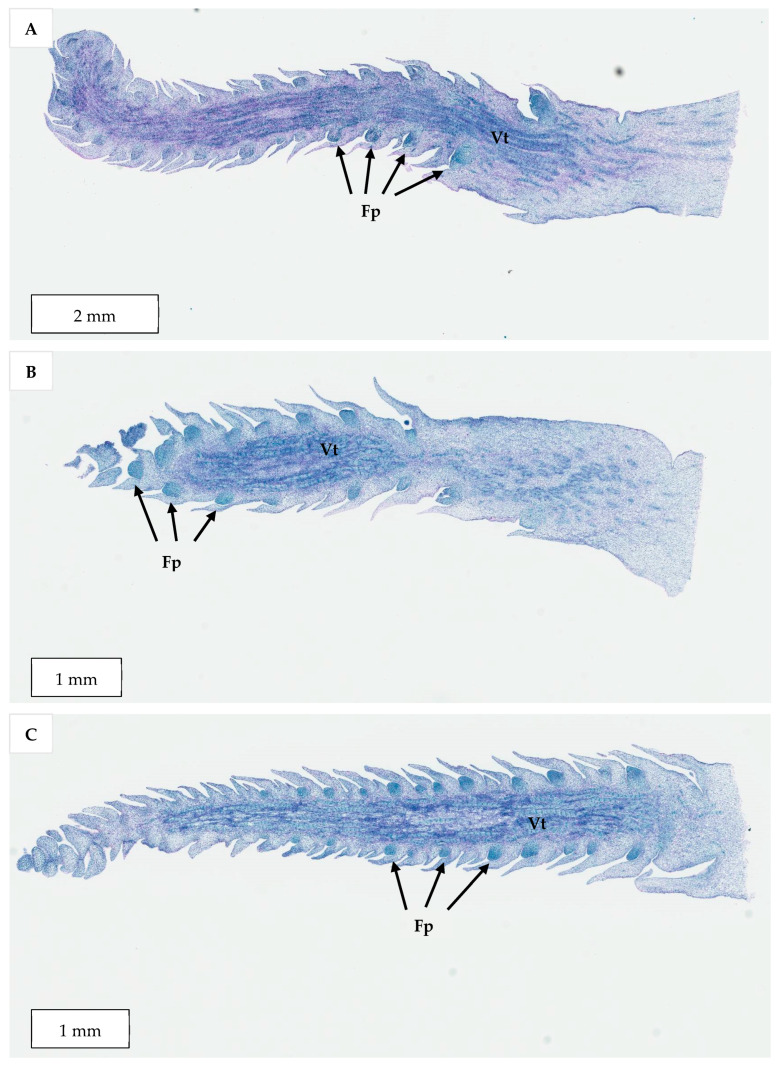
A series of longitudinal sections taken through an inflorescence rachilla from (**A**) −3, (**B**) −4, and (**C**) −5 developmental stages with the presence of floral primordia (Fp) and vascular tissues (Vt).

**Figure 15 plants-14-03123-f015:**
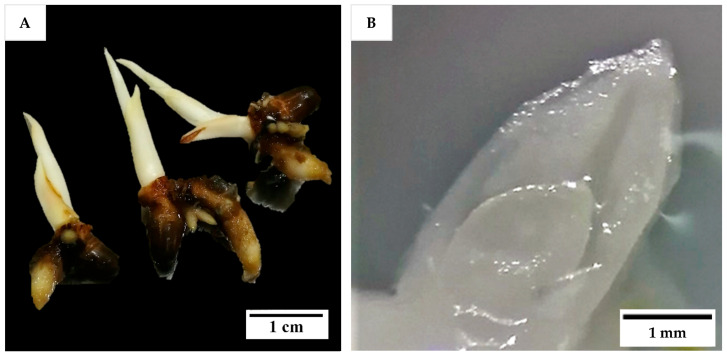
(**A**) In vitro-grown 3-month-old seedlings used for meristem isolation. The shoot was excised under a clean dissecting microscope to obtain (**B**) a shoot tip, each containing a meristem, which was cultured onto a shoot induction medium.

**Figure 16 plants-14-03123-f016:**
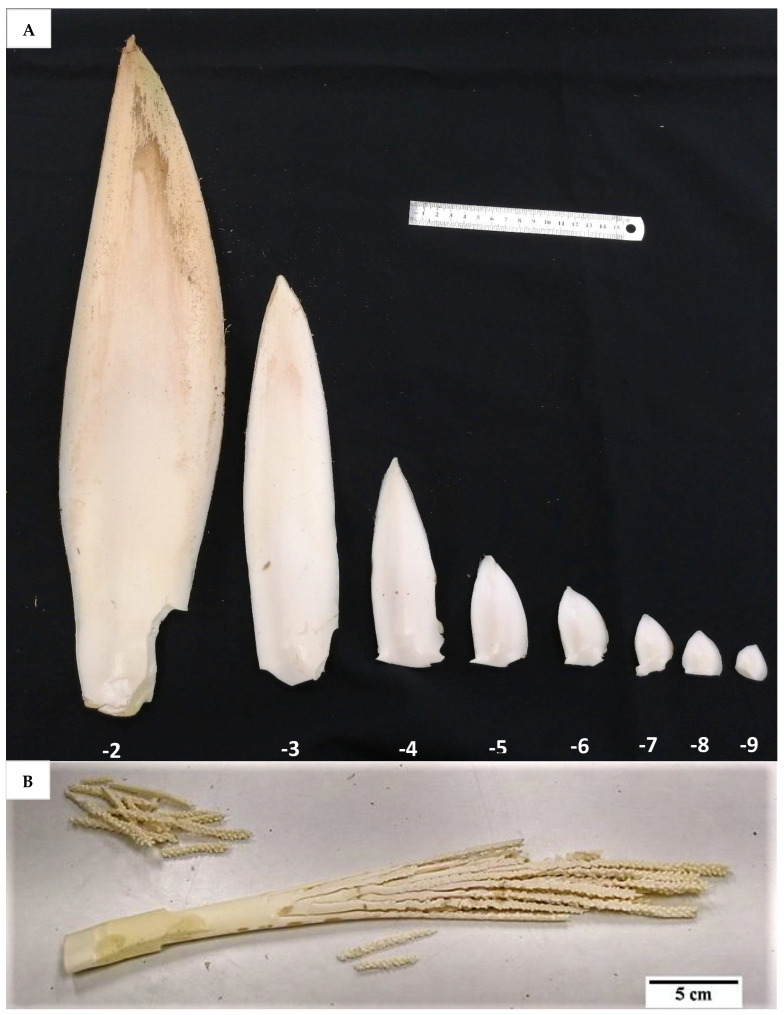
(**A**) A series of inflorescence explants (Australian coconut variety, CAIRNS) enclosed in their double spathe with decreasing developmental stages from ca. 2 to 9 months before opening (−2 to −9), representing different maturity or age stages of development, and (**B**) the rachillas isolated from −2 developmental stage after removing the two covering spathe layers.

**Figure 17 plants-14-03123-f017:**
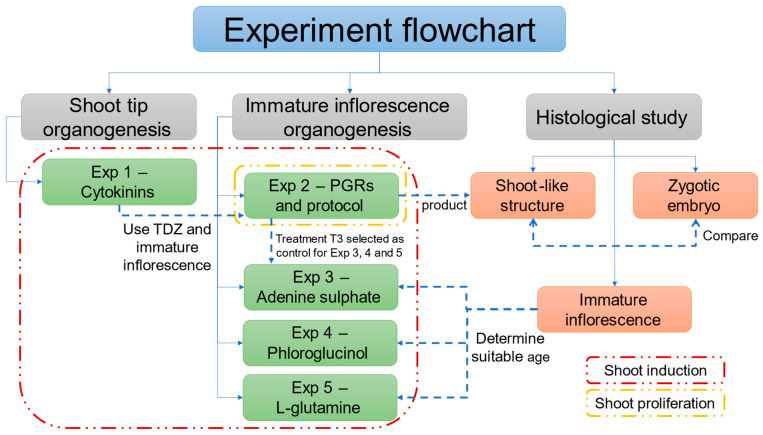
A flowchart showing the links between the experiments conducted in this study showing the type of treatment and the explants used in the shoot induction and proliferation stages of direct shoot organogenesis in coconut. Key: PGR: Plant growth regulator.

## Data Availability

Data is contained within the article or Appendix A.

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
