# Peer review of "Initiation of Direct Shoot Organogenesis in Coconut Using Immature Inflorescence"

_plants, 2025, doi:10.3390/plants14203123_

Round 1
Reviewer 1 Report
Comments and Suggestions for Authors
The current paper described direct plant regeneration of coconut plants from immature explants. The authors clearly demonstrated that explants rich in auxin production can serve as excellent starting points for de novo shoot formation.
The authors do a good work, but paper organization require significant improvements.
Please, come carefully through text, avoid grammar mistake, adjust scientific sense. Please, do not forget that organogenesis regulated by endogenous hormones and ion balance is a key for hormone production.
Minor comments.
Line 24: = exogenous plant growth regulators
Line 27: “in all treatments consisted of plant growth regulators except for those without activated charcoal.” = in all treatments consisted of exogenous plant growth regulators except (reduction in exogenous regulators and rising local temperature by activated charcoal).
Line 35: “improvements which warrants further studies for potential improvement” please, slightly edit.
Line 59: “Plant growth regulators (PGRs) have shown an important role in coconut organogenesis” = Exogenous plant growth regulators (EPGRs) have shown an effective inducer of exogenous auxin and it canalization as mechanism of de novo shoot induction. https://doi.org/10.3390/ijpb16030097
Line 65: “maximum 2 shoots produced” ¿? Per what? Per plants/explants? Which explants?
Line 72: “of several cytokinins, auxins and an auxin uptake inhibitor” – exogenous??
Line 75: “Coconut immature inflorescence tissues remain the preferable explant tissue” = Coconut immature inflorescence tissues remain the preferable explant tissue because of higher activity of endogenous auxin biosynthesis,
Line 92: “the rate or shoot = the rate of shoot”
Line 107: “also control hyperhydricity” ¿?
Line 134: “organogenesis was in the cytokinins experiment”??
Line 171: “Effects of PGRs and protocols on organogenesis from inflorescence sections” ?? exogenous PGR?? Effect of protocols??
Lines 175- 181: these results, most probably, because of “Y3 medium” with unsuitable nutrient balance.
I would suggest re-designing medium composition for next experiments. Y3 medium is full of chloride, indeed, which may one of the ley factors of the process authors observed.
Lines 190-195: good results!
Line 201: it is better to move table 1 immediately after first mentioned.
Line 209:it will be useful to remove hyperhydricity by balancing medium in next experiments.
Lines 331-333: “Cytokinin’s such as BAP and thidiazuron (TDZ) have been found to induce direct shoot formation from coconut vegetative shoots or inflorescence tissues (Raju, 2006; Vidhanaarachchi and Weerakoon, 1997).” = cytokinin itself did not induce shoots de novo. Please, add number as citation, not only authors.
Line424; “ascorbic acid (200 mg L−1) “ ??? Ascorbic acid cannot be add to the medium because of ASC instability and rapid conversion to oxalic acid. Authors can add to discussion that observed effect is not ASC, but oxalic acid. Golubitskii, G. B., Budko, E. V., Basova, E. M., Kostarnoi, A. V., & Ivanov, V. M. (2007). Stability of ascorbic acid in aqueous and aqueous-organic solutions for quantitative determination. Journal of Analytical Chemistry, 62(8), 742-747.
Line 525- 528: SIM/SMM seems to be confusing. Please, provide clear composition. Make table 1 clearer.
Line 556: “The results (Section 3.6) suggested immature inflorescences at the -4 and developmental stages were the most suitable for shoot induction via. organogenesis. “ = results, not M&M.
Comments on the Quality of English Languagemany corrections are required as grammar.
Author Response
Comment 1: The current paper described direct plant regeneration of coconut plants from immature explants. The authors clearly demonstrated that explants rich in auxin production can serve as excellent starting points for de novo shoot formation.
The authors do a good work, but paper organization require significant improvements.
Please, come carefully through text, avoid grammar mistake, adjust scientific sense. Please, do not forget that organogenesis regulated by endogenous hormones and ion balance is a key for hormone production.
Response 1: Thank you for the positive comments and constructive feedback. We have revised the manuscript and made improvements on the organization, grammar and scientific sense throughout the manuscript as suggested. We have carefully reviewed and performed the changes for each comment.
Comment 2: Minor comments.
Line 24: = exogenous plant growth regulators
Response 2: We have added “Exogenous” to Line 24 for clarity as suggested.
Comment 3: Line 27: “in all treatments consisted of plant growth regulators except for those without activated charcoal.” = in all treatments consisted of exogenous plant growth regulators except (reduction in exogenous regulators and rising local temperature by activated charcoal).
Response 3: Authors thank the reviewer for suggesting to include proposed explanation of activated charcoal in the abstract, however we prefer not to include because the data do not directly support conclusions on reduction of exogeneous plant growth regulators or changes in local temperature by activated charcoal. In order to maintain accuracy and avoid speculation, we have restricted our reporting to the observed results. We agree that these mechanisms are interesting possibilities to why this is happening in the treatment with activated charcoal, but they are beyond the scope of the present study. However, we have added “exogeneous” to the plant growth regulators (Line 27) as suggested.
Comment 4: Line 35: “improvements which warrants further studies for potential improvement” please, slightly edit.
Response 4: Thank you for pointing out and it has been changed to “ several improvements which warrants further studies to develop …” (Line 35).
Comment 5: Line 59: “Plant growth regulators (PGRs) have shown an important role in coconut organogenesis” = Exogenous plant growth regulators (EPGRs) have shown an effective inducer of exogenous auxin and it canalization as mechanism of de novo shoot induction. https://doi.org/10.3390/ijpb16030097
Response 5: Thanks for the additional information and this has been added (Line 58-61) with a little modification based on the literature provided by reviewer.
Comment 6: Line 65: “maximum 2 shoots produced” ¿? Per what? Per plants/explants? Which explants?
Response 6: We have made the changes accordingly, please refer to Line 65.
Comment 7: Line 72: “of several cytokinins, auxins and an auxin uptake inhibitor” – exogenous??
Response 7: The word “Exogenous” has been added for clarity (Line 71-72).
Comment 8: Line 75: “Coconut immature inflorescence tissues remain the preferable explant tissue” = Coconut immature inflorescence tissues remain the preferable explant tissue because of higher activity of endogenous auxin biosynthesis,
Response 8: We thank the reviewer for this suggestion. However, the cited references do not provide evidence for higher endogenous auxin biosynthesis in immature inflorescence tissues. To our knowledge, no published work is directly supporting this claim in coconut. Therefore, we retain the original statement that immature inflorescence tissues remain the preferable explant due to the ease of collection and ability to produce true-to-type clones when compared to zygotic sources, without speculating on the underlying mechanisms.
Comment 9: Line 92: “the rate or shoot = the rate of shoot”
Response 9: We have changed for better clarity to “used to promote shoot induction…” (Line 92) as suggested.
Comment 10: Line 107: “also control hyperhydricity” ¿?
Response 10: Yes, this is correct based on the literature, shown in the abstract. (https://doi.org/10.1007/s11627-013-9491-2)
Comment 11: Line 134: “organogenesis was in the cytokinins experiment”??
Response 11: Thanks for pointing out, we have changed accordingly to provide clarity (Line 133-134).
Comment 12: Line 171: “Effects of PGRs and protocols on organogenesis from inflorescence sections” ?? exogenous PGR?? Effect of protocols??
Response 12: Thanks for the comment. This has been changed to “Effects of exogenous PGRs based on protocols on…” (Lines 171, 530) for clarification.
Comment 13: Lines 175- 181: these results, most probably, because of “Y3 medium” with unsuitable nutrient balance.
I would suggest re-designing medium composition for next experiments. Y3 medium is full of chloride, indeed, which may one of the ley factors of the process authors observed.
Response 13: Thank you for the valuable feedback. Y3 medium was chosen because multiple in vitro studies on coconut have reported it as the most effective medium. However, as reviewer has suggested, it is possible that nutrient balance influenced the observed results, indicating that further optimization will be necessary for future experiments.
Comment 14: Lines 190-195: good results!
Response 14: We thank the reviewer for their positive comment and appreciation of this study.
Comment 15: Line 201: it is better to move table 1 immediately after first mentioned.
Response 15: We sincerely thank the reviewer for this suggestion. However, we consider that the details presented in Table 1 are more appropriately placed in the Methodology section rather than in the Results. In line with the Plants journal guidelines, where the Methodology follows after the Results and Discussion, we have retained the current placement and clarified that Table 1 can be found in Section 4.3.
Comment 16: Line 209:it will be useful to remove hyperhydricity by balancing medium in next experiments.
Response 16: Yes, this is an important observation that requires further optimization, and we highlighted it in Lines 411–413 as a key area for future research. Balancing the nutrient composition of the medium is one of the important factors that could reduce hyperhydricity.
Comment 17: Lines 331-333: “Cytokinin’s such as BAP and thidiazuron (TDZ) have been found to induce direct shoot formation from coconut vegetative shoots or inflorescence tissues (Raju, 2006; Vidhanaarachchi and Weerakoon, 1997).” = cytokinin itself did not induce shoots de novo. Please, add number as citation, not only authors.
Response 17: Thank you for the comment and we agree that existing shoot or floral meristems are usually involved to induce shoots in coconut. The citation has been included (Line 338).
Comment 18: Line424; “ascorbic acid (200 mg L−1) “ ??? Ascorbic acid cannot be add to the medium because of ASC instability and rapid conversion to oxalic acid. Authors can add to discussion that observed effect is not ASC, but oxalic acid. Golubitskii, G. B., Budko, E. V., Basova, E. M., Kostarnoi, A. V., & Ivanov, V. M. (2007). Stability of ascorbic acid in aqueous and aqueous-organic solutions for quantitative determination. Journal of Analytical Chemistry, 62(8), 742-747.
Response 18: We thank the reviewer for the suggestion. To clarify, ascorbic acid was not added to the culture media, and it was used only in the liquid medium during shoot tip excision. All shoot tips were treated identically during excision and then transferred to different culture media, so any observed differences are solely due to the media treatments. Therefore, we believe it is beyond the scope to discuss potential effects of ascorbic acid degradation or oxalic acid formation in the manuscript.
Comment 19: Line 525- 528: SIM/SMM seems to be confusing. Please, provide clear composition. Make table 1 clearer.
Response 19: SIM and SMM have been expanded as shoot induction media and shoot multiplication media respectively to provide clarity (Line 537-538). Table 1 header has been changed to medium and culture conditions and the key for each abbreviation can be found below the table (Line 561-563), and the treatment separation is now presented clearly.
Comment 20: Line 556: “The results (Section 3.6) suggested immature inflorescences at the -4 and developmental stages were the most suitable for shoot induction via. organogenesis. “ = results, not M&M.
Response 20: Thanks for pointing out and this has been removed from Materials and Methods as suggested (Line 566).
Reviewer 2 Report
Comments and Suggestions for Authors
This is a comprehensive and well-conducted study with new results for the development of direct shoot organogenesis in palm species that provides insights for coconut cultivation. This study aims to establish the initial stages of in vitro propagation of coconut plants. The final stages are still pending until the seedlings are acclimated in the greenhouse and soil.
Include data comparing direct shoot organogenesis with somatic embryogenesis for producing coconut plants; which of the two would be a faster and more economical method as an alternative clonal propagation route to contribute to the generation of elite coconut planting material.
Author Response
Comment 1: This is a comprehensive and well-conducted study with new results for the development of direct shoot organogenesis in palm species that provides insights for coconut cultivation. This study aims to establish the initial stages of in vitro propagation of coconut plants. The final stages are still pending until the seedlings are acclimated in the greenhouse and soil.
Include data comparing direct shoot organogenesis with somatic embryogenesis for producing coconut plants; which of the two would be a faster and more economical method as an alternative clonal propagation route to contribute to the generation of elite coconut planting material.
Response 1: We thank the reviewer for the encouraging feedback. While we recognize the importance of comparing direct shoot organogenesis and somatic embryogenesis in terms of efficiency and cost, this analysis is beyond the scope of the present study as the pathway to obtain complete plants via direct shoot organogenesis is still under development. We agree this is an important aspect, and it could be a valuable direction for future work once the system is fully established.
Reviewer 3 Report
Comments and Suggestions for Authors
Dear Authors, please consider the following suggestion:
Why did you limit the regeneration time for inflorescences in Exp. 1 to 4 weeks, but significantly extend it in Exp. 2? It was obvious that this type of explant would not regenerate shoots within 4 weeks. So what was the point of using media with different concentrations of BAP and TDZ? In Fig. 3B, you only demonstrated that different concentrations of BAP and TDZ did not negatively affect explant survival. And did you have any reason to expect that the result would be different in the presence of cytokinins? The explants for this experiment, as you indicate, were obtained in vitro, so only the method of explant isolation could have affected their viability. If you noticed, as you describe in Figure 2C, an increase in inflorescence structures - lines 152-153, why didn't you extend the experimental time or include BAP and TDZ in Experiment 2?
You report that the best results for experiment 2 were obtained with the T7 (SIM) medium. Why did you use the T3 SIM medium in subsequent experiments?
You wrote "intact inflorescences were collected from a known Indonesian (Banyumas Tall; BT) and an unknown Australian coconut variety (referred to as CAIRNS)" but you did not include the results for both varieties.
Furthermore, you report the influence of genotype on the obtained results (line 392). Be cautious with such a statement, as you analyzed various genotypes in subsequent stages of the experiment: Nias Yellow Dwarf, Banyumas Tall; BT), an unknown Australian variety, and Sampoorna variety, but you didn't compare them.
Additional comments in the manuscript.

Author Response
Comment 1: Dear Authors, please consider the following suggestion:
Why did you limit the regeneration time for inflorescences in Exp. 1 to 4 weeks, but significantly extend it in Exp. 2? It was obvious that this type of explant would not regenerate shoots within 4 weeks. So what was the point of using media with different concentrations of BAP and TDZ?
Response 1: To clarify, Experiment 1 was purposely conducted for a short period to screen which explant type (shoot tip or inflorescence) is responsive to cytokinins. Although inflorescence sections are unlikely to regenerate shoots in four weeks, testing both BAP and TDZ allowed us to identify preliminary responsiveness before committing to longer, more resource-intensive culture periods in Experiment 2, where extended culture time was applied to evaluate shoot induction more thoroughly.
Comment 2: In Fig. 3B, you only demonstrated that different concentrations of BAP and TDZ did not negatively affect explant survival. And did you have any reason to expect that the result would be different in the presence of cytokinins? The explants for this experiment, as you indicate, were obtained in vitro, so only the method of explant isolation could have affected their viability. If you noticed, as you describe in Figure 2C, an increase in inflorescence structures - lines 152-153, why didn't you extend the experimental time or include BAP and TDZ in Experiment 2?
Response 2: Yes, higher concentrations of cytokinins can negatively affect plant growth, so it was important to test a range of concentrations. The short duration of Experiment 1 is explained in Response 1, and TDZ was included in Experiment 2 with extended culture periods. Furthermore, literature on date palm, oil palm, and coconut shows that combinations of cytokinins and auxins are often required for optimal shoot induction. Therefore, we tested multiple hormone combinations, including BAP with other hormones, to identify the most effective treatment rather than using BAP alone.
Comment 3: You report that the best results for experiment 2 were obtained with the T7 (SIM) medium. Why did you use the T3 SIM medium in subsequent experiments?
Response 3: We appreciate the reviewer’s comment. This is because T3 recorded a high shoot induction percentage (75%) and a comparable number of shoot-like structures per explant, with a lower standard error (5.6 ± 0.6), making it the frontrunner. Therefore, the PGR combination used in T3 was applied to subsequent experiments (Experiments 3–5). This explanation has been added in Lines 196–200.
Comment 4: You wrote "intact inflorescences were collected from a known Indonesian (Banyumas Tall; BT) and an unknown Australian coconut variety (referred to as CAIRNS)" but you did not include the results for both varieties.
Response 4: This has been clarified in Line 437 to 439: BT was used for Experiment 1, while CAIRNS was used for Experiments 2–5. Therefore, the results for both varieties are included in the study.
Comment 5: Furthermore, you report the influence of genotype on the obtained results (line 392). Be cautious with such a statement, as you analyzed various genotypes in subsequent stages of the experiment: Nias Yellow Dwarf, Banyumas Tall; BT), an unknown Australian variety, and Sampoorna variety, but you didn't compare them.
Response 5: We thank the reviewer for this comment. Since all genotypes are not used in all experiments, authors intentionally avoided such comparison. To clarify, Nias Yellow Dwarf and BT were used in Experiment 1, Sampoorna was used for histology analysis, and the unknown Australian variety (CAIRNS) was used in Experiments 2–5. The statement regarding genotype influence was based on observations from CAIRNS alone, where the control treatments across Experiments 3–5 used different CAIRNS palms and showed a range of results. We have revised the wording in Line 398-399 to avoid overstating.
Comment 6: Additional comments in the manuscript.
Response 6: All additional revisions have been made as suggested. The word ‘tissues’ has been replaced with ‘explants’ throughout the manuscript where appropriate. Lines 28–31 have been revised for clarity, the caption of Figure 7 has been updated, and ‘in vitro’ in Line 437 has been removed as recommended.

Round 2
Reviewer 1 Report
Comments and Suggestions for Authors
Thank you for very professional response and clear answer. As general suggestions, in the future try to based y0ur medium composition on the role of each ions and adjust accordingly.
For the current text on line 206 I see: "see Table 1 in Section 4.3)" but table 1 is only on like 569. I agree with you that table 1 fit with M&M, but in this case reader should "jump" to line 569 to evalute graphs...
My best regards!
Author Response
Comment 1: Thank you for very professional response and clear answer. As general suggestions, in the future try to based y0ur medium composition on the role of each ions and adjust accordingly.
For the current text on line 206 I see: "see Table 1 in Section 4.3)" but table 1 is only on like 569. I agree with you that table 1 fit with M&M, but in this case reader should "jump" to line 569 to evalute graphs...
My best regards!
Response 1: We thank the reviewer for the valuable suggestions and constructive feedback, which will be considered in future experiments. Table 1 has been moved to the Results Section 2.2 (Line 211-216) as suggested to improve the flow and readability of the manuscript.